# On the Morphological Deviation in Additive Manufacturing of Porous Ti6Al4V Scaffold: A Design Consideration

**DOI:** 10.3390/ma15144729

**Published:** 2022-07-06

**Authors:** Seyed Ataollah Naghavi, Haoyu Wang, Swastina Nath Varma, Maryam Tamaddon, Arsalan Marghoub, Rex Galbraith, Jane Galbraith, Mehran Moazen, Jia Hua, Wei Xu, Chaozong Liu

**Affiliations:** 1Institute of Orthopaedic & Musculoskeletal, Division of Surgery & Interventional Science, University College London, Royal National Orthopaedic Hospital, Stanmore, London HA7 4LP, UK; seyed.naghavi.14@ucl.ac.uk (S.A.N.); haoyu.wang@ucl.ac.uk (H.W.); t.varma@ucl.ac.uk (S.N.V.); m.tamaddon@ucl.ac.uk (M.T.); 2Department of Mechanical Engineering, University College London, London WC1E 7JE, UK; arsalan.marghoub.15@ucl.ac.uk (A.M.); m.moazen@ucl.ac.uk (M.M.); 3Department of Statistical Science, University College London, London WC1E 6BT, UK; r.galbraith@ucl.ac.uk (R.G.); j.galbraith@ucl.ac.uk (J.G.); 4School of Science and Technology, Middlesex University, London NW4 4BT, UK; j.hua@mdx.ac.uk; 5National Engineering Research Center for Advanced Rolling and Intelligent Manufacturing, Institute of Engineering Technology, University of Science and Technology Beijing, Beijing 100083, China; weixu@ustb.edu.cn

**Keywords:** additive manufacturing, geometry deviation, mechanical properties, nanoindentation, surface roughness, Ti6Al4V scaffolds, bone scaffolds

## Abstract

Additively manufactured Ti scaffolds have been used for bone replacement and orthopaedic applications. In these applications, both morphological and mechanical properties are important for their in vivo performance. Additively manufactured Ti6Al4V triply periodic minimal surface (TPMS) scaffolds with diamond and gyroid structures are known to have high stiffness and high osseointegration properties, respectively. However, morphological deviations between the as-designed and as-built types of these scaffolds have not been studied before. In this study, the morphological and mechanical properties of diamond and gyroid scaffolds at macro and microscales were examined. The results demonstrated that the mean printed strut thickness was greater than the designed target value. For diamond scaffolds, the deviation increased from 7.5 μm (2.5% excess) for vertical struts to 105.4 μm (35.1% excess) for horizontal struts. For the gyroid design, the corresponding deviations were larger, ranging from 12.6 μm (4.2% excess) to 198.6 μm (66.2% excess). The mean printed pore size was less than the designed target value. For diamonds, the deviation of the mean pore size from the designed value increased from 33.1 μm (−3.0% excess) for vertical struts to 92.8 μm (−8.4% excess) for horizontal struts. The corresponding deviation for gyroids was larger, ranging from 23.8 μm (−3.0% excess) to 168.7 μm (−21.1% excess). Compressive Young’s modulus of the bulk sample, gyroid and diamond scaffolds was calculated to be 35.8 GPa, 6.81 GPa and 7.59 GPa, respectively, via the global compression method. The corresponding yield strength of the samples was measured to be 1012, 108 and 134 MPa. Average microhardness and Young’s modulus from α and β phases of Ti6Al4V from scaffold struts were calculated to be 4.1 GPa and 131 GPa, respectively. The extracted morphology and mechanical properties in this study could help understand the deviation between the as-design and as-built matrices, which could help develop a design compensation strategy before the fabrication of the scaffolds.

## 1. Introduction

The titanium alloy (Ti6Al4V) is a two-phase alloy (α + β) that has attracted great attention in the aerospace, marine, chemical and biomedical industries [1]. This is mainly due to its excellent physical and mechanical properties, as well as the proper balance of processing characteristics, including high strength and ductility at low temperatures, low density, excellent corrosion-resistant properties, high fracture toughness, good plastic workability, heat treatability and weldability [2,3,4].

Many researchers have performed experimental and numerical investigations on the mechanical behaviour of Ti6Al4V alloy deformation at macro and micro/nanoscales using traditional compression tests and conventional nanoindentation tests [2,5,6,7]. In recent years, nanoindentation techniques have been widely used to characterize the local micromechanical properties of materials at micro and nanoscales. These include microhardness, Young’s modulus, yield stress, work-hardening exponents, and the indentation size effect (dependence of hardness on indentation depth) [8]. All these mechanical properties are obtained by the load–displacement curves (P–h), which are generated via indenting the material as either a load or depth control [9]. As an alternative to traditional testing, the nanoindentation method has some major benefits. For example, only a small amount of material is required for these techniques; they are nondestructive, relatively fast, simple and inexpensive to perform and provide precise measurements and control over loading rates and indentation depth during indentation [9].

Additive manufacturing (AM) has been advancing over the past 25 years, bridging many gaps in developing new products that conventional manufacturing methods were limited in producing. In recent years, AM has been used in many high-tech industries, including biomedical, aerospace, marine and automotive [10,11]. AM gives any user the freedom of designing three-dimensional (3D) models with complex geometries, consolidating them into one part and fabricating them [12]. Other benefits of AM over the conventional manufacturing method are rapid prototyping, lightweight design and functional integration [13]. There are several factors that can affect the quality of AM products; these include raw material properties, design strategies (consolidation, orientation, topology and geometrical optimization), manufacturing process and postprocessing [13].

It has been shown that additively manufactured Ti6Al4V alloys can reduce mechanical properties (fatigue life, stiffness and strength) to approximately 90% of the expected values of traditionally manufactured Ti6Al4V [14,15]. This is due to the highly localized heat input and short interaction time, large temperature gradients and high cooling rates that are present. These unique thermal features dramatically affect as-built microstructures and lead to high residual stresses in AM-fabricated Ti6Al4V products, which in turn affect their macroscopic performances [16]. In addition, the inevitably formed defects during the AM processes significantly compromise the products’ mechanical properties. Zhou et al. [17] showed that by post-heat treating the AM samples, residual stresses could be relieved, which can significantly improve the fatigue strength of specimens. It is also known that the fatigue strengths of traditionally manufactured parts are higher than AM parts due to a smoother surface finish [18].

Uncontrolled micropores are generally developed in the process of AM. These are the gas pores that are created due to the trapped gas inside the hollow powders which did not escape during the solidification process and have a diameter of approximately 1–100 μm [19,20,21]. Another type of micropores is called lack-of-fusion pores, which are generated due to unoptimized melting conditions such as insufficient laser energy trying to melt an excessive amount of powders, which results in inadequate melting and weak bonding between layers. Lack-of-fusion pores tend to be larger than gas pores. During loading, the sharp edges of these micropores act as nucleation sites for adiabatic shear bands and initiate microcracks, which results in premature failure [14,22]. Some studies have shown a significant reduction (approximately 25% of the expected values) in Young’s modulus and yield strength of AM products, raising concerns about the manufacturing methods and process (SLM, EBM and DED) used to produce Ti6Al4V alloy products [23].

The fabrication of AM scaffolds results in irregular and inhomogeneous strut thicknesses and surface topologies. The surface roughness of these scaffolds has been shown to have a major influence on the material’s functional properties, such as frictional behaviour [24], fluid dynamics [25], heat transfer [26], optical and mechanical properties [27,28] and biological cell attachment onto the surface of the scaffold [29]. These irregularities can cause local stress, which results in early crack initiation and growth, and significantly decreases the fatigue properties of the scaffold [30]. Many studies have investigated the effect of surface roughness on osseointegration, long-term bone–implant fixation and removal torque required to detach the bonding [31,32,33]. Ronold et al. [34] showed that tensile strength is proportional to surface roughness up to 3.9 μm, and a further increase in roughness results in reduced tensile strength. A review paper by Wennerberg et al. [29] has shown that the optimal surface roughness for enhanced bone–implant attachment is between 1 and 2 μm. Surface roughness is affected by several factors, which are stability, dimensions, the behaviour of the melt pool during the SLM process and the orientation of the strut to the built plate [35]. It is known that struts printed in the vertical direction (θ = 0) to the built plate have a smoother surface finish than horizontal struts (θ = 90). This is because when printing a strut facing upwards, it has direct contact with the laser beam, hence, the majority of the powders are melted [36]. However, as the angle of the printing sturt and the built plate increases (θ = 0→90), the amount of unsintered powders increases [37]. This is because the horizontal struts have the potential to overheat, which can cause partially fused powder particles to adhere to them and result in a lower surface quality [38].

Several postprocessing treatments can reduce the surface roughness of the as-built samples, including electropolishing, chemical process and mechanical polishing. Chemical etching is an acid-based solution that penetrates the pores of the scaffold and dissolves the irregularities of the surfaces of the struts, which results in a reduced surface roughness [39]. By controlling the chemical composition and etching time of the process, the amount of reduced surface roughness can be controlled [40]. Mechanical polishing includes sandblasting, which is a process where aluminium oxide particles are sprayed onto the surface of the scaffold at a high speed and remove any partially bonded powders, leaving a smoother surface roughness [41]. This process also causes severe local plastic deformations, which result in the strain hardening of the surface of the scaffold and increased stiffness, strength and fatigue performance [42].

With currently available AM technology, we often see a geometrical deviation between the as-designed and as-built porous structures. This deviation is further increased by reducing the pore size [43]. This geometrical deviation indeed affects the expected mechanical properties of the structure and may result in closed pores with trapped powders [44], as well as geometry. It has been reported that the strut thickness is a function of the angle at which the material is formed on the building plate [44,45]. As the angle of vertically printed struts (0 degrees) approaches that of horizontal struts (90 degrees), the amount of excess thickness between the as-designed and as-built structures increases. To achieve a “precision” and satisfactory in vivo performance of orthopaedic implants, understanding the process capability of additive manufacturing is an important issue, especially for porous implants that require good bone ingrowth for a stable mechanical fixation. Gignato et al. [46] performed a study on improving the accuracy of SLM-manufactured parts. They showed that regardless of the printing angle of struts (from 0 to 90 degrees), the geometrical accuracy would not exceed 15 μm in a standard deviation, and sloped planes at 15–20 degrees should preferably be avoided to achieve dimensional accuracy. Two significant factors that affect the geometrical accuracy of the manufactured parts are the building angle and the type of geometry that is being fabricated [46]. Bageri et al. [47] investigated developing a compensation strategy to reduce the geometry mismatch of a Tetrahedron-based structure. They used the results of an error analysis of a planer sample (spider-web) design with strut orientations from 0 to 90 degrees at intervals of 30 degrees, to develop a thickness compensation expression as a function of the angle of printing the strut with respect to the built plate. They managed to reduce the percentage error of the horizontal struts between the as-designed and as-built structures from 60% to 3.1%. In practice, there are two factors that can potentially limit the application of the AM technique for the purposes of orthopaedic implants: one is the need for a controlled geometry and architecture, and the other is the requirement for the printed implants to be mechanically vigorous. Therefore, in order to manufacture clinically useful orthopaedic implants through the effective operation of AM technology, it is important to understand the manufacturability of the AM systems [48]. To the best of the authors’ knowledge, there has been no investigation on understanding the deviation of vertical and horizontal printing struts between as-deigned and as-built TPMS gyroid and diamond scaffolds.

In this study, morphological deviations between the as-built and as-designed AM fabricated porous Titanium scaffolds, for both diamond and gyroid structures, were investigated. The effect on the mechanical behaviour of the AM Ti6Al4V alloy at macro and micro/nanoscale was examined and analysed. The obtained results have the potential to guide the design and manufacture of porous orthopaedic implants using additive manufacturing technology.

## 2. Materials and Methods

### 2.1. Powder Material

Ti6Al4V-grade 23 ELI powder supplied by A GE Additive Company (AP&C, Saint-Eustache, QC, Canada) and manufactured by Darwin Health Technology Co. (Guangzhou, China) was used to fabricate the porous scaffolds. Morphology of the powder particles was examined using scanning electron microscopy (SEM) (Thermo/FEI Quanta 200F, Thermo Fisher Scientific, Waltham, MA, USA). As can be seen in Figure 1, the powder particles had a nearly spherical shape with very smooth surfaces, indicating a good flowability of the particles. The particle size distribution (ASTM B417) and apparent density (ASTM B822) of the Ti6Al4V powder are shown in Table 1. The chemical composition of this Ti6Al4V powder was also investigated (ASTM B348) and is shown in Table 2. As illustrated, Ti6Al4V contained a very low level of carbon, oxygen, iron and nitrogen.

### 2.2. Design and Manufacturing of Porous Scaffolds

Two triply periodic minimal surface (TPMS) structures, i.e., gyroid (G) and diamond (D) scaffolds, were developed by using nTopology platform (version 3.25.3, New York, NY, USA) (Figure 2) [49]. The equations used to develop the Schoen gyroid and Schwartz diamond scaffold are shown below [50]:

Schoen gyroid unit cell:(1)$$\emptyset_{G}\left(x,y,z\right)=\sin\left(\frac{2\pi }{a}x\right)\cos\left(\frac{2\pi }{b}y\right)+\sin\left(\frac{2\pi }{b}y\right)\cos\left(\frac{2\pi }{c}z\right)+\sin\left(\frac{2\pi }{c}z\right)\cos\left(\frac{2\pi }{a}x\right)=R$$

Schwartz diamond unit cell:(2)$$\begin{array}{ll} \emptyset_{D}\left(x,y,z\right) & =\sin\left(\frac{2\pi }{a}x\right)\sin\left(\frac{2\pi }{b}y\right)\sin\left(\frac{2\pi }{c}z\right)\\ & +\sin\left(\frac{2\pi }{a}x\right)\cos\left(\frac{2\pi }{b}y\right)\cos\left(\frac{2\pi }{c}z\right)+\cos\left(\frac{2\pi }{a}x\right)\sin\left(\frac{2\pi }{b}y\right)\cos\left(\frac{2\pi }{c}z\right)\\ & +\cos\left(\frac{2\pi }{a}x\right)\cos\left(\frac{2\pi }{b}y\right)\sin\left(\frac{2\pi }{c}z\right)=R \end{array}$$
where (x,y,z) are the Cartesian coordinate system and a, b, c are the length of the unit cell in x, y and z direction. In this study, a, b and c were kept constant to obtain isotropic properties. The constant R is the defined relative density.

Sheet thickness and porosity of the TPMSs were kept constant at 300 μm and 62%, respectively, for both gyroid and diamond scaffolds. Pore size was defined as the interconnected pore size, which is the diameter of a sphere that passes through the largest pore of the porous structure. In this study, only two pore sizes were selected for morphological, mechanical and nanoindentation investigation. These were sheet TPMS gyroid unit cells with a pore size of 800 μm (G800) and sheet TPMS diamond unit cells with a pore size of 1100 μm (D1100). The cylindrical test specimens for G800 and D1100 were ∅11.04 × 16.56 mm and ∅14.055 × 21.083 mm, respectively (Figure 2). These specimens were selected and manufactured with SLM for morphological and mechanical investigation.

All samples were manufactured using an SLM machine (EOS M280, Krailling, Germany), from Ti6Al4V (grade 23 ELI) alloy. Printing parameters were optimised by Darwin Health Technology Co. (Guangzhou, China) to gain the highest quality of print with the least amount of deviation between the as-designed and as-built scaffolds. The details of the laser parameters are outlined in Table 3. The fabricated samples were then removed from the build plate with a wire cutting machine, and air was blown to remove any unmelted powder. To enhance the mechanical properties of the lattice structure, the samples were then thermal treated at 820 °C with a heating rate of 9 °C/min for 2 h and then cooled to room temperature in a furnace. Sandblasting with quartz sand with a particle size of 50 µm at a pressure of 0.6 MPa was used to further remove the loosely bonded particles.

To examine and ensure reproducibility of the data, five replicate samples of each porous scaffolds, G800 and D1100, were manufactured and analysed.

### 2.3. Morphological and Microstructure Characterisations

All fabricated samples were scanned and their morphologies were characterised using a SkyScan (model 1172, Bruker, Billerica, MA, USA) high-resolution microcomputed tomography (micro-CT) scanner. The scans were performed with a tube voltage of 102 kV, tube current of 96 µA, a scan time of 30 min and a voxel size of 10 × 10 × 10 µm. Each sample was rotated from 0° to 180° in steps of 0.5°, and 5 images were recorded to obtain an average radiograph image. Micro-CT data were then reconstructed into 2D slices, representing the cross-sectional images of the scaffolds with a commercial software package (NRecon, Skyscan N.V., Kontich, Belgium). The reconstruction process included beam hardening correction of 35%, ring artifact reduction of 10, and lower and upper histogram ranges of 0 and 0.15, respectively.

For each replicate structure, strut thickness (ST) and pore size (PS) were measured for vertical struts at different locations within 4 slices, which were vertical cross-sections equally spaced across the structure. In addition, ST and PS were similarly measured for horizontal struts in 4 other slices, also equally spaced across the structure. Within each slice, ST was measured at 30 locations and PS was measured at another 30 locations. Thus, for a given structure type, there were 20 slices showing vertical struts (4 slices in each of 5 replicates) with 30 ST measurements and 30 PS measurements in each slice, and 20 slices showing horizontal struts with 30 ST measurements and 30 PS measurements in each slice. ImageJ software package (National Institutes of Health, Bethesda, Rockville, MD, USA) was used to measure the ST and PS across the samples.

The surface roughness of the specimens was measured using a 4K digital microscope (VHX-7000, Keyence, Osaka, Japan) with 500 magnification. Morphology of the as-built and sandblasted samples were observed using a scanning electron microscopy (SEM) using a Thermo/FEI Quanta 200F machine with 5 kV voltage and spot size of 2, while samples were being mounted on a stub.

### 2.4. Statistical Analysis

Numerical and graphical analyses of strut thickness and pore size measurements were performed in R (R Foundation for Statistical Computing, Vienna, Austria). These included calculations of group means and standard deviations, estimates with confidence intervals, analyses of variance with standard diagnostics and calculation of components of variance.

### 2.5. Mechanical Properties

#### 2.5.1. Macroscale Mechanical Properties (Global Mechanical Tests)

To obtain the mechanical properties of the samples, compressive and tensile test samples were tested in compression (ISO 17340-2014) [51] and tension (ISO 6892:2019) [52], respectively, using an Instron mechanical testing machine (model 5985, 250 kN load cell, Instron, Norwood, MA, USA) (Figure 3). Compression samples were placed between two flat hard metal plates and only vertical movement with a constant strain rate of 0.01 mm s^−1^ was allowed until failure. Bulk tensile samples were fixed with a grip length of 20 mm and were tested in tension with a constant strain rate of 0.01 mm s^−1^ until failure. Same ISO standards were used to determine the compressive and tensile stiffness (E) and yield strength ($\sigma_{y}$) of each sample from the obtained stress–strain curve. Stiffness was measured as the maximum slope of the elastic region of the stress–strain curve. Yield strength ($\sigma_{y}$) was computed by intersecting the stress–strain curve with a 0.2% offset line parallel to the elastic region. Compressive stiffness was calculated as follows:(3)$$\mathrm{Stiffness}\;\left(E\right)=\frac{\sigma }{\varepsilon }=\frac{\frac{F}{A}}{\frac{\Delta L}{L}}\;$$
where σ is the stress, F is the vertical reaction force, A is the initial cross-sectional area of the upper surface of the cylinder, ε is the strain, ΔL is the displacement of the upper surface in the vertical direction and L is the initial length of the scaffold. Tensile stiffness was calculated as follows:(4)$$\mathrm{Stiffness}\;\left(E\right)=\frac{\sigma }{\varepsilon }=\frac{\frac{F}{A_{o}}}{\frac{\Delta L}{L_{c}}}$$
where $A_{o}$ is the initial cross-sectional area of the sample within the gauge length ($L_{o}$) and $L_{c}$ is the initial length between the grips.

#### 2.5.2. Nanoindentation Sample Preparation

Gyroid and diamond scaffolds were embedded in epoxy resin (EpoThin, Buehler, Lake Bluff, IL, USA) and were kept inside a vacuum chamber at 25 Bar pressure for 30 min to be able to remove any trapped bubbles within the porous structure of the scaffolds. The resin was cured by placing the embedded specimens inside a pressure curing chamber at 2 MPa for a day. Cured samples were sectioned at the continuous visible struts level for further analysis. The samples were polished with sandpaper grits of P400, P800, P1200 and P2500 and a ChemoMet polishing cloth with polishing alumina until roughness of below 1 μm was obtained.

#### 2.5.3. Nanoscale Mechanical Properties

The micromechanical analysis was performed using a nanoindenter (Anton Paar GmbH, Graz, Austria) equipped with a diamond Berkovich indenter (3-sided pyramid) with Young’s modulus and Poisson’s ratio of 1141 GPa and 0.07, respectively. The nanoindentation tests were conducted considering a reduced modulus of elasticity, hardness and contact depth and using Oliver and Pharr method [53]. Mechanical properties investigated in this work were local microhardness (H) and Young’s modulus (E) of the material. Two loads of 50 and 100 mN were used to record the load–displacement curves (P–h curve) with a constant loading unloading rate of 100 mN/min. A total of 49 indentations was performed on the diamond scaffold and 24 indentations on the gyroid scaffold.

A typical schematic and load–displacement (P–h) curve of an indented sample is shown in Figure 4. The diamond indenter was inserted into the surface of the material causing a deformation on the surface, reaching a certain maximum depth ($h_{m}$) and contact depth ($h_{c}$). Once the indenter was removed, some of the deformed material was restored, reaching the final depth ($h_{f}$). $h_{m}$ at maximum load ($P_{\max}$) could be obtained from this P–h curve. The Young’s modulus and hardness values were calculated from the $P_{\max}$, initial unloading slop S and $h_{m}$, which was dependent on the elasticity of the material. According to Kick’s law, the loading curve could be expressed by the following equation:(5)$$P=Ch^{2}$$
where P is the indenter load, h is the indenter displacement and C is the loading curvature depending on the elastic–plastic material properties, as well as indenter geometry.

Based on Oliver and Pharr’s model [53], the unloading curve could be expressed as:(6)$$P=B\times \left(h-h_{f}\right){\;}^{m}$$
where B and m are fitting parameters and $h_{f}$ is the residual depth after unloading. Microhardness H could be calculated as follows:(7)$$H=\frac{P_{\max}}{A_{ef}}$$
where $A_{ef}$ is the effective contact area, which was shown by:(8)$$A_{ef}=Kh_{c}{}^{2}$$
where K is a constant, whose value was 24.56 for the Berkovich indenter in this case, and $h_{c}$ is the measured contact depth in the unloading curves [54].

Elastic contact stiffness, S, is the unloading slope at the maximum indentation depth $h_{m}$, and is shown by the following equation:(9)$$S={\frac{dP}{dh}|}_{h=h_{m}}=m\times B\left(h_{m}=h_{f}\right){\;}^{m-1}$$

Contact depth, $h_{c}$, was estimated with the following equation:(10)$$h_{c}=h_{m}-\gamma \frac{P_{\max}}{S}\;$$
where γ is the tip-dependent geometry factor, which was equal to 0.75 for a Berkovich indenter in this case [55].

Young’s modulus, E, could be expressed as:(11)$$E=\left(1-v^{2}\right){\left(\frac{1}{E^{*}}-\frac{1-v_{i}{}^{2}}{E_{i}}\right)}^{-1}$$
(12)$$E^{*}=\sqrt{\frac{\pi }{2\lambda }}\times \frac{S}{\sqrt{A}}\;$$
where v is the Poisson’s ratio of the specimen, which was 0.33 for the Ti6Al4V alloy, $E_{i}$ and $v_{i}$ are Young’s modulus and the Poisson’s ratios of the diamond indenter whose values were 1141 GPa and 0.07, respectively [54]. $E^{*}$ is the reduced Young’s modulus, λ, is a correction factor associated with the indenter shape, which was approximately 1.05 for the Berkovich indenter [56].

## 3. Results and Discussion

### 3.1. Morphology of Porous Biomaterials: Designed vs. Manufactured

Key morphological characteristics such as the strut thickness (ST) and pore size (PS) of the manufactured samples were characterised and measured by a micro-CT and were compared to their designed target values. ST and PS were measured based on vertical struts (0 degrees) and horizontal struts (90 degrees) as shown in Figure 5. There were five replicate structures of each type (diamond and gyroid), and within each replicate, measurements of ST and PS were determined at 30 different locations in each of the four slices. The full set of measurements is shown in the form of dot plots in Appendix A). We found that the printed struts tended to be thicker than the target value, and more so for horizontal struts, which in turn corresponded to a smaller printed pore size compared to the target and more so for horizontal struts.

#### 3.1.1. Variation of Individual Measurements

The means and standard deviations of the 30 measurements of ST and PS in each slice are listed in Appendix A, and Figure 6 shows a plot of these standard deviations against the mean excess (slice means minus their target values). The variation of individual measurements within each slice was comparable across the 20 slices within each group. The pooled standard deviations for all slices within each group (each with 20 × 29 = 580 degrees of freedom) are presented in Table 4, denoted by s1. These were all of a similar magnitudes (between 17.3 and 19.5 microns), except for horizontal struts in gyroids, which were approximately twice as large (33.8 microns for PS and 37.1 microns for ST). They represented how much individual measurements in the same slice varied about their slice means. The standard deviations (s2) in Table 4 describe how much individual measurements in a replicate structure varied about the replicate mean. They were larger than s1, because the slice means also varied. Details of how they were calculated are given in the Appendix A. Histograms of the 600 pooled differences of the measurements from the mean slice for each group, along with fitted normal distributions, are shown in Appendix A. The empirical distributions tended to be more peaked (leptokurtic) with longer tails compared with a normal distribution. This may have had implications for the performance of the structure.

#### 3.1.2. Variation of Slice Means

Estimates of excess from the target value and comparisons between orientations and structure types were appropriately based on an analysis of variation of the slice means, which were shown as red triangles in Appendix A and listed in Appendix A. Figure 7 shows, for each structure type and strut orientation, the mean ST plotted against the mean PS for each of the four slices within each of the five replicates (i.e., 20 points in each panel). The horizontal red lines indicate the target values for ST and the vertical red lines indicate the target values for PS. The points were negatively correlated (as might be expected) and the correlation was stronger for vertical struts than for horizontal struts. The grey lines indicate where the sum of ST and PS would equal the sum of their target values. The axis scales were designed to allow comparisons between panels, as well as within panels.

In every case, the mean pore size was below the target value, albeit by differing amounts. The discrepancy from the target was greater for the horizontal struts, compared with the vertical ones, and, furthermore, that difference was more pronounced for gyroids than for diamonds. For the mean strut thickness, all of the points were above the target value of 300 microns, except for vertical struts in diamonds, where 5 of the 20 points were below it. In that case, a formal test of the null hypothesis that the true mean ST equalled 300 microns gave a *p*-value of 0.0061, indicating strong evidence that the true mean was greater than 300 microns. The excess thickness above the target was much larger for the horizontal struts, compared with the vertical ones, and, again, this difference was more pronounced for gyroids compared with diamonds. For each combination of structure type, variable and strut orientation, we carried out a two-way analysis of variance (ANOVA) of the 20 slice means, where the factors were replicates (with levels 1, 2, 3, 4 and 5) and the slice order (with levels 1, 2, 3 and 4). In all cases, there was very little evidence of any systematic variation with the slice order. Indeed, there was very little evidence of systematic variation between the replicates either, except for the horizontal struts within gyroids, where there was some evidence that the replicate means varied for both ST and PS measurements. We, therefore, calculated the between-slice variances of the slice means from the one-way ANOVA of slices within replicates, each with 15 degrees of freedom. The resulting standard deviations are given in Table 4, along with the corresponding mean values (i.e., the means of the 20 slice means), the target values and estimates of the excess (mean minus target in microns) and relative excess (mean minus target as a percentage of the target) with 95% confidence intervals. The *t*-test and 95% confidence intervals formally assumed that the slice means were normally distributed. The Q–Q plots in Appendix A and a related discussion in the Appendix A confirmed that this was reasonable. It is known in the literature that as the printed strut orientation changes from vertical (0 degrees) to horizontal (90 degrees), the excess thickness between the as-designed and as-built increases. If so, the strut thickness would be a function of the angle to the building plate [44,45]. This is because the vertical struts can self-support themselves while being printed; however, in horizontal struts, the amount of partially molten powder particles increases on the downward surface and enhances this excess thickness [57]. Percentage excess values in Table 4 also confirmed this, where we had less than 5% error in the vertical struts for both diamond and gyroid scaffolds, which were in less bad agreement with the target values. However, in horizontal struts, gyroid scaffolds had a larger ST percentage error (66.2%) when compared to diamond scaffolds, which were approximately 35.1%. To reduce the deviations between the as-designed and as-built structures, some preventive and postprocessing actions could be taken, such as design compensation strategies before printing, optimising machine parameter tuning, chemical etching, electropolishing and sandblasting [44,58].

### 3.2. Surface Roughness (As-Built and Sandblasted)

Figure 8a,b show the as-built morphology of the fabricated gyroid and diamond scaffolds. As it can be seen, the surface roughness of the scaffolds was reduced significantly after post-treating them with sandblasting (Figure 8c,d). To quantify this numerically, surface roughness was measured with a digital microscope, showing that the as-built (Figure 8e) and sandblasted scaffolds (Figure 8f) had an average and standard deviation of 16.98 (6.06) μm and 1.21 (0.62) μm, respectively. The obtained average surface roughness was low enough to lie within the suggested limit (1–2 μm) to have an enhanced osseointegration and long-term bone–implant fixation [29]. It is worth mentioning that even though we had a smooth surface on the outer surface of the scaffolds, we noticed that as we sliced through the sample, the surface roughness started to increase, suggesting that the sandblasting particles did not manage to reach the inner sections of the scaffold due to the small design of the pore size.

### 3.3. Macroscale Mechanical Properties (Global Mechanical Tests)

Figure 9a shows the true stress–true strain graph of the bulk compression and dogbone tension sample. Compressive Young’s modulus (E) and yield strength ($\sigma_{y}$) (0.2% offset) were measured to be 35.8 ± 0.7 GPa and 1012 ± 15 MPa, respectively. Tensile Young’s modulus (E) and yield strength ($\sigma_{y}$) (0.2% offset) were measured to be 95.1 ± 0.9 GPa and 788 ± 7 MPa, respectively. The expected Young’s modulus and yield strength of Ti6Al4V were approximately 110 GPa and 880 MPa, respectively. We can see that the measured Young’s modulus in the additively manufactured compression and tension samples was 67% and 14% lower than the expected value, and that the yield strength of the compression and tension sample was within 15% of the expected value. Garciandia showed a similar result with a 3D-printed bulk Ti6Al4V, having a Young’s modulus between 26 and 42 GPa [23]. There are several reasons which could help us understand why the experimental Young’s modulus of the compression sample was only approximately 33% of the expected value. This includes the development of microporosity while printing the samples, which could lead to the partial bonding of the particles within the structure, lowering the global stiffness of the sample. After scanning the sample with micro-CT, we could see some visible micropores mostly on the outer edge of the fabricated sample. To understand the quality of the printed Ti6Al4V in a micro/nanoscale, a nanoindentation study was performed and the results were compared with similar studies in the literature. Figure 9b shows the stress–strain curve of the G800 and D1100 scaffolds, where G800 had a stiffness of 6.81 GPa and yield strength of 108 MPa. D1100 had a greater stiffness at 7.59 GPa and yield strength of 134 MPa. This indicated that at the same porosity level of 62%, the diamond scaffold was stiffer by approximately 11% and stronger by approximately 24% than the gyroid scaffold.

### 3.4. Nanoindentation Results

#### 3.4.1. Visual Inspection

An optical image of indentations determined under a load of 100 mN is shown in Figure 10. The indentation marks were visible on the struts of the scaffolds. However, the microstructure of α and β phases was not clearly visible under the digital microscope. Results of load–displacement (P–h) could give us a suggestion on where the indentations were created in an α or β phase.

#### 3.4.2. Load–Displacement (P–h) Curves for Ti6Al4V

Figure 11 shows the loading–displacement (P–h) curves, which were generated by the Berkovich indenter on the surface of the polished scaffolds under the load control mode. In total, 24 and 49 indentations were created on the gyroid and diamond scaffolds, respectively. Each P–h curve contained three sections, the loading, dwell time where the maximum loading occurred and unloading. As we can see, for a constant applied load of 100 mN, the amount of indentation depth varied with every indentation that was created, which was marked by a different colour. From left to right, the depth increased where smaller depths corresponded to the indentations that were created in the α-phase, which were harder, and higher depths were responsible for indentations that were created in the α + β phases or β phase alone, which were softer. This was also reported in the literature [2,59]. It was also found that the P–h curves with different depths showed a similar shape, which confirmed the repeatability of the indentation tests.

#### 3.4.3. Nano Hardness and Young’s Modulus

Hardness and Young’s modulus were calculated by the formulas mentioned in Section 2.5.3. Figure 12a shows the measured hardness with respect to the indentation depth. It is clear that the hardness decreased gradually as the penetration depth increased. The results showed a decreasing exponential trend; however, there was certainly a limit where the hardness value stabilized as the penetration depth increased. In this study, we did not reach any stabilized hardness limit. At a 50 mN load for the diamond scaffold, the maximum hardness (5.6 GPa) was at the depth of 536 nm, which was 75% higher than the lowest hardness (3.2 GPa) at a depth of 729 nm. At a 100 mN load for the diamond scaffold, the maximum hardness (5.0 GPa) was at the depth of 816 nm, which was 79% higher than the lowest hardness (2.8 GPa) at a depth of 1114 nm. At a 100 mN load for the gyroid scaffold, the maximum hardness (5.1 GPa) was at the depth of 798 nm, which was 200% higher than the lowest hardness (1.7 GPa) at a depth of 1417 nm. The average hardness was measured to be 4.1 ± 0.1 GPa. Figure 12e shows the comparison of the hardness measurements conducted with nanoindentation tests in this study with previous references for Ti6Al4V [2,6,7,8,9,59,60,61]. We can see that our hardness values were in agreement with the literature. For example, Wen et al. showed a hardness range of 4.3–6.2 [59]. Cai et al. showed a hardness variation from 4.0 to 5.5 GPa [7]. Han et al. showed a hardness range of 4.09–4.71 GPa [6]. The variation of the hardness results was due to the indentation on different phases of the Ti6Al4V, where indentations in the α phase resulted in a larger hardness value.

Figure 12c shows the measured Young’s modulus with respect to the indentation depth. Similar to the hardness results, the measured Young’s modulus decreased gradually with an expected exponential decay as the penetration depth increased. At a 50 mN load for the diamond scaffold, the maximum Young’s modulus (151 GPa) was at the depth of 536 nm, which was 34% higher than the lowest Young’s modulus (113 GPa) at a depth of 729 nm. At a 100 mN load for the diamond scaffold, the maximum Young’s modulus (152 GPa) was at the depth of 833 nm, which was 42% higher than the lowest Young’s modulus (107 GPa) at a depth of 1050 nm. At a 100 mN load for the gyroid scaffold, the maximum Young’s modulus (149 GPa) was at the depth of 798 nm, which was 84% higher than the lowest Young’s modulus (81 GPa) at a depth of 1417 nm. The average Young’s modulus was measured to be 131 ± 2 GPa. Figure 12f shows the comparison of the Young’s modulus measurements conducted using nanoindentation tests in this study, with previous references for Ti6Al4V [2,7,8,9,59,60]. We can see that our Young’s modulus values were in agreement with the literature. For instance, Wen et al. showed a Young’s modulus range of 120–147 [59]. Haghshenas et al. showed a Young’s modulus range of 130–150 GPa [9]. Cai et al. showed a Young’s modulus variation from 110 to 112 GPa [7]. Similar to the hardness results, the variation of the Young’s modulus results was due to the influence of different indentation sites (α and β phases) of the Ti6Al4V, where indentations in the α phase resulted in a greater Young’s modulus value. The average values of hardness and Young’s modulus were from the average values of all α and β phase measurements.

## 4. Conclusions

In the present study, the morphological deviation, surface roughness treatment and material mechanics of additively manufactured Ti6Al4V alloy scaffolds were investigated via micro-CT, digital microscope, nanoindentation and conventional compression methods. The following conclusions could be drawn:The morphological deviation of strut thicknesses and pore sizes in AM-manufactured porous scaffolds varied within a structure, with a standard deviation of approximately 20 μm, except for the horizontal struts in gyroids, where the standard deviation was larger (approximately 36 μm). The mean printed strut thickness was greater than the designed value. For the diamond scaffolds, the difference increased from 7.5 to 105.4 μm when the strut orientation changed from vertical to horizontal. For gyroids, the corresponding deviations were larger, ranging from 12.6 to 198.6 μm, correspondingly. The mean printed pore size was less than the designed pore size and was negatively correlated with the mean strut thickness, though the two means did not add to the sum of their targets. For diamonds, the difference in mean pore size from the target increased from 33.1 μm for vertical struts to 92.8 μm for horizontal struts. The corresponding increase for gyroids was larger, ranging from 23.8 to 168.7 μm.Postprocessing the scaffolds with sandblasting could reduce the surface roughness of the as-built scaffolds from 16.98 ± 0.51 μm to 1.21 ± 0.05 μm. The obtained surface roughness would lie within the suggested value from the literature for enhanced osseointegration and long-term bone–implant fixation.Via the conventional compression mechanical test (macroscale), Young’s modulus and yield strength of the bulk additively manufactured Ti6Al4V were measured to be 35.8 ± 0.7 GPa and 1012 ± 15 MPa, respectively. Where Young’s modulus was only approximately 33% of the expected value (110 GPa), the yield strength was 15% higher than the expected value (880 MPa). For the gyroid scaffold, Young’s modulus and yield strength were 6.81 GPa and 108 MPa, respectively. Corresponding values for diamond scaffold were 7.59 GPa and 134 MPa.An indentation size effect was clearly observed in the hardness and Young’s modulus versus depth graphs, where hardness and Young’s modulus were gradually decreased by increasing the indentation depth. Average Young’s modulus was calculated to be 131 ± 2 GPa, which was in good agreement with the reported Young’s modulus for the Ti6Al4V alloy in the literature (112–151 GPa). Average hardness was measured to be 4.1 ± 0.1 GPa, which was in good agreement with the reported hardness for the Ti6Al4V alloy in the literature (3.6–5.2 GPa).The extracted morphology and mechanical properties in this study could help understand the deviation between the as-design and as-built matrices, which could help develop a design compensation strategy before the fabrication of the TPMS gyroid and diamond scaffolds.

## Figures and Tables

**Figure 1 materials-15-04729-f001:**
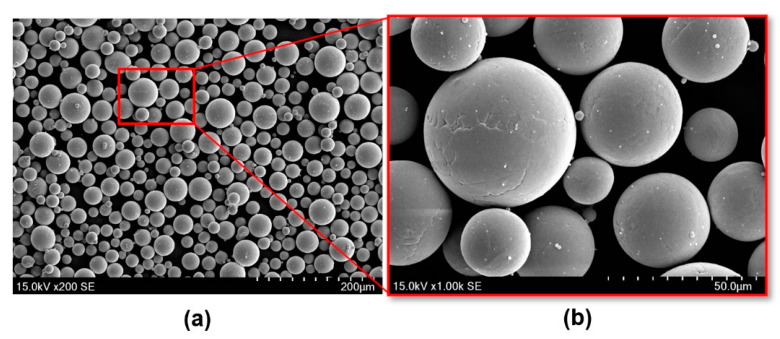
SEM image showing the morphology of spherical Ti6Al4V raw powder used for manufacturing lattice structures via SLM (**a**) and powder’s surface appearance (**b**).

**Figure 2 materials-15-04729-f002:**
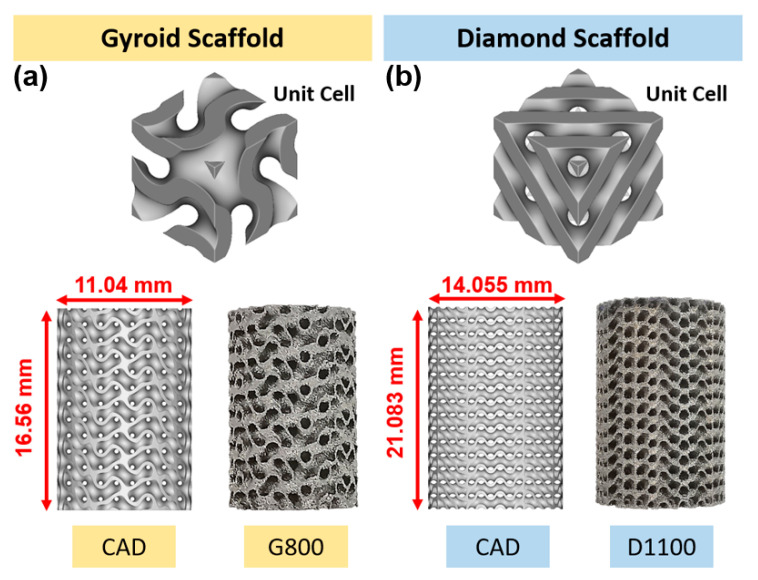
Three-dimensional-printed samples: (**a**) gyroid scaffold; (**b**) diamond scaffold.

**Figure 3 materials-15-04729-f003:**
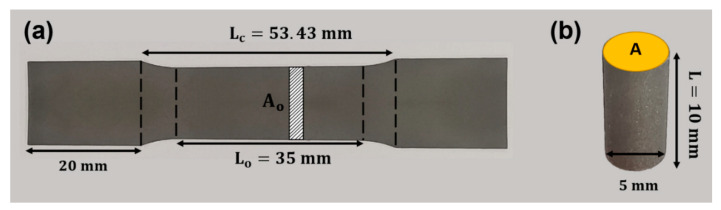
Dimensions of additively manufactured Ti6Al4V (**a**) tension dogbone sample (**b**) compression bulk sample.

**Figure 4 materials-15-04729-f004:**
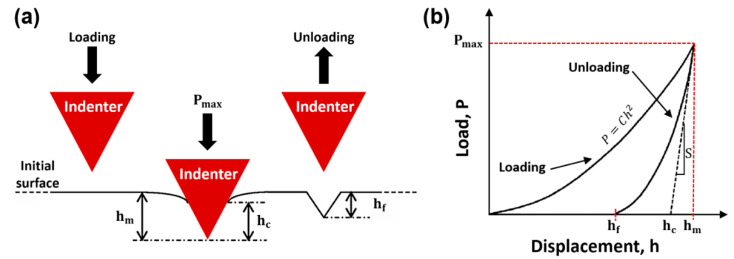
(**a**) Schematic representation of the interaction between Berkovich indenter tip and the sample (Oliver and Pharr, 1992): Pmax—maximum applied load; $h_{m}$—maximum depth; $h_{c}$—contact depth; $h_{f}$—final depth. (**b**) A typical load–displacement indentation curve.

**Figure 5 materials-15-04729-f005:**
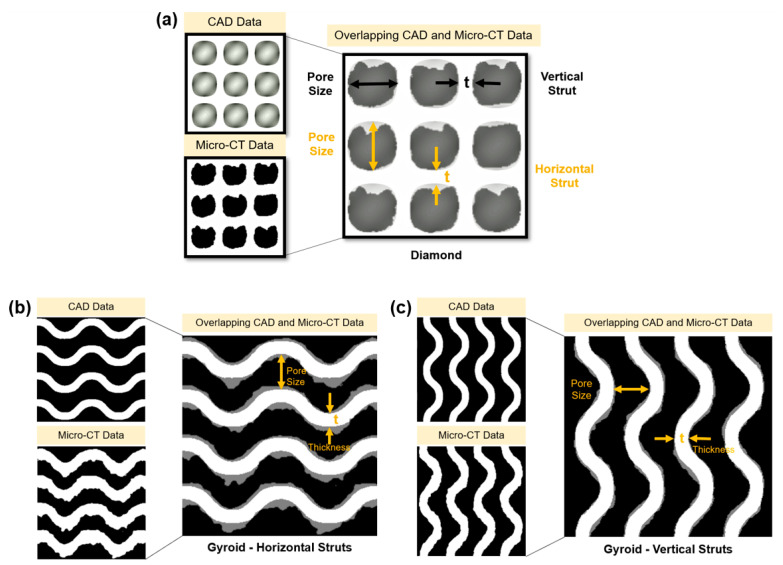
Overlapping CAD and micro-CT morphology data in vertical and horizontal struts orientations for (**a**) diamond scaffold, (**b**) gyroid (Horizontal Struts) scaffold and (**c**) gyroid (Vertical Struts) scaffold.

**Figure 6 materials-15-04729-f006:**
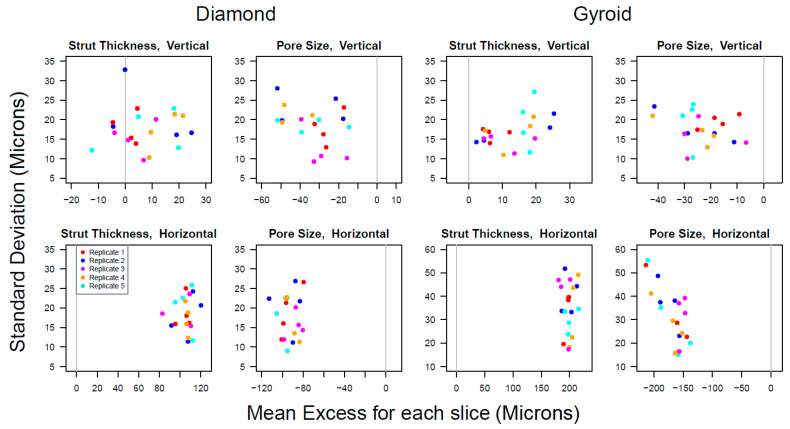
Within-slice standard deviations were plotted against the mean excess (i.e., the slice mean minus target value) for each structure type, variable and strut orientation. Different colours denote the 5 different replicates of each printed structure. The *x*-axis scales differ in each panel. The *y*-axis scales are the same in all panels, except for the horizontal strut measurements for gyroids, where they cover a wider range. The grey vertical line at zero marks where the mean measurement would equal the target value. Replicates 1, 2, 3, 4, 5 are shown by red, blue, purple, orange and turquoise colours, respectively.

**Figure 7 materials-15-04729-f007:**
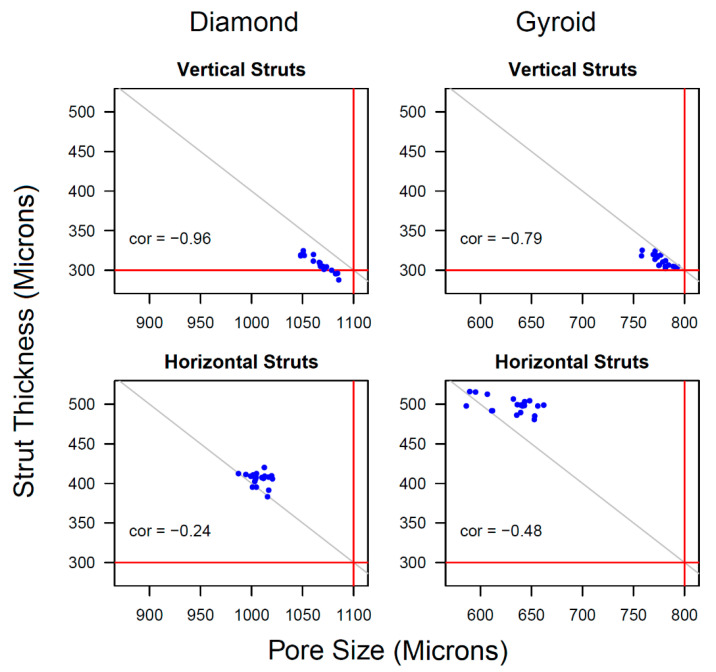
Mean strut thickness plotted against mean pore size for each slice within each replicate structure, for each combination of structure type and strut orientation. There are 20 points in each panel, each being the mean of the 30 measurements in the same slice. The red lines show the target values for each variable. The grey line in each panel is given by the equation x + y = tS + tP, where tS and tP are the target values for strut thickness and pore size, respectively. The same axis scales were used in all panels, except that the location of the pore size scale, which differed for diamonds and gyroids because of their different target values.

**Figure 8 materials-15-04729-f008:**
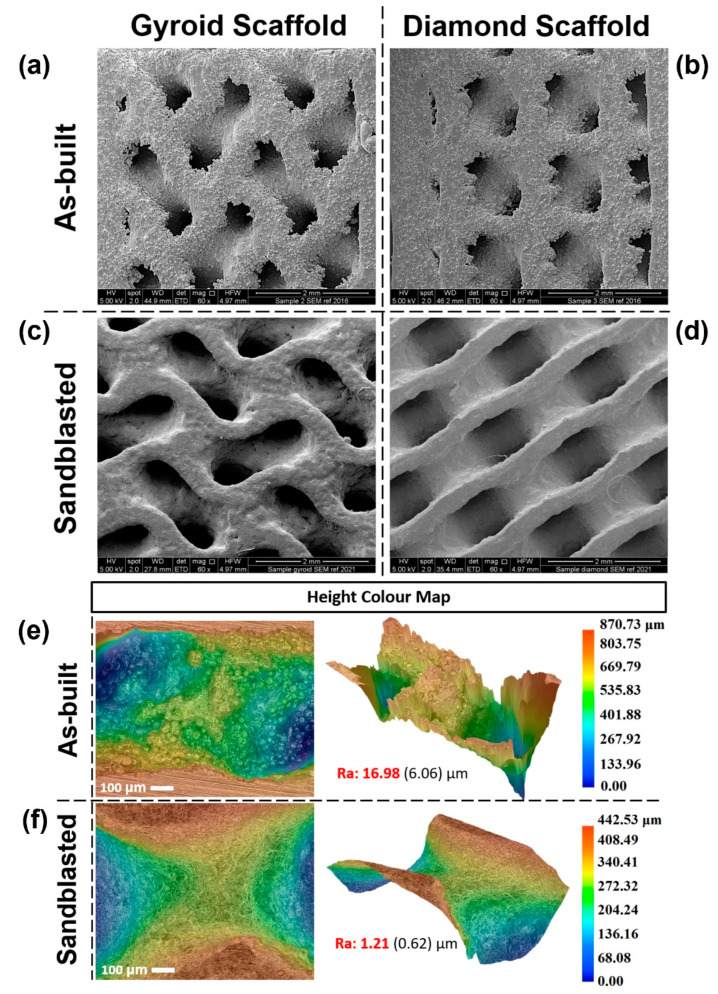
SEM images showing the morphology of the printed Ti6Al4V samples: (**a**) as-built gyroid, (**b**) as-built diamond, (**c**) sandblasted gyroid and (**d**) sandblasted diamond. Digital microscope images showing the morphology of the printed Ti6Al4V samples: (**e**) as-built, (**f**) sandblasted.

**Figure 9 materials-15-04729-f009:**
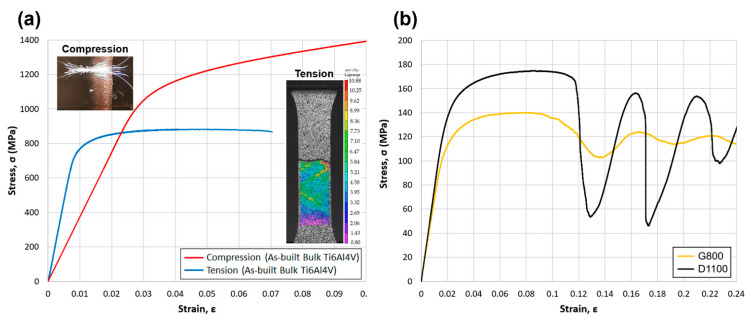
(**a**) Compression and tension stress–strain behaviour of bulk additively manufactured Ti6Al4V. Colour scale shows the surface Lagrange strain (**b**) Compression stress–strain behaviour of G800 and D1100 scaffolds.

**Figure 10 materials-15-04729-f010:**
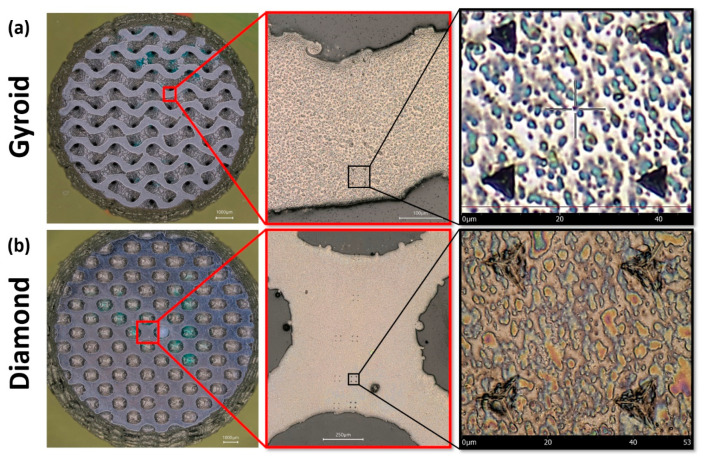
Optical image of indentations created under a load of 100 mN on (**a**) gyroid scaffold and (**b**) diamond scaffold.

**Figure 11 materials-15-04729-f011:**
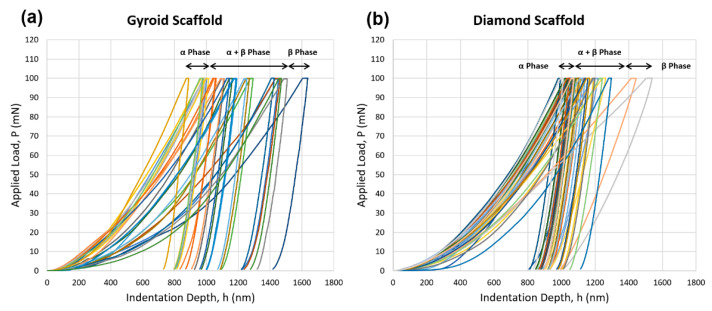
Indentation loading–unloading curves (or P–h curves) performed at 100 mN on (**a**) gyroid scaffold and (**b**) diamond scaffold. Different colours of the curves show different indentation loading–unloading that were performed on the gyroid and the diamond scaffold.

**Figure 12 materials-15-04729-f012:**
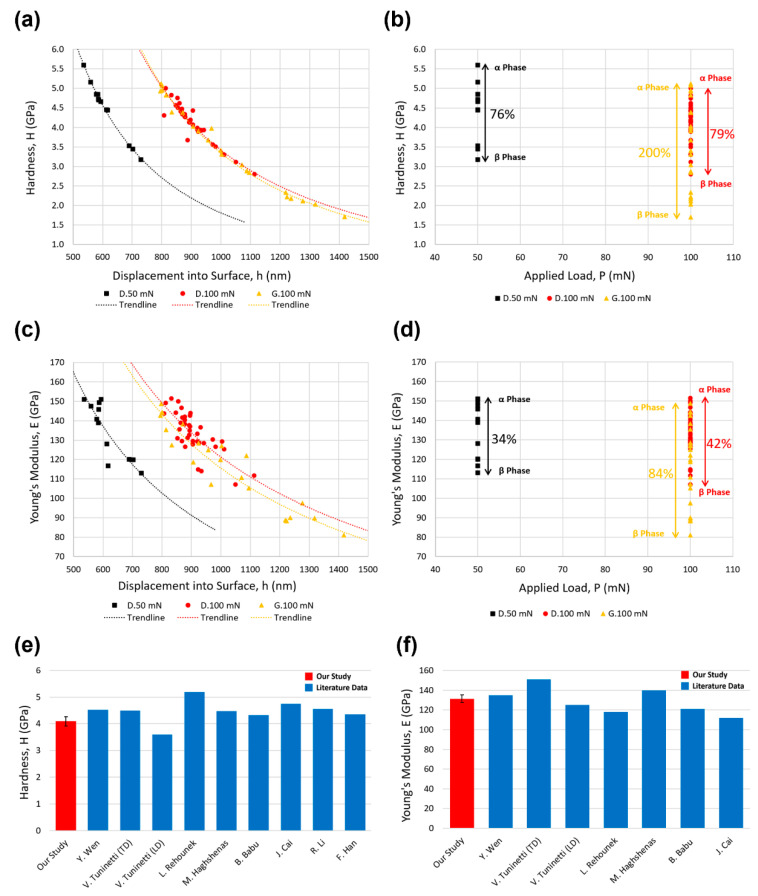
Measured hardness with respect to (**a**) displacement into the surface and (**b**) applied load. Measured Young’s modulus with respect to (**c**) displacement into the surface and (**d**) applied load. Comparison of measured (**e**) hardness and (**f**) Young’s modulus with the literature. The error bars denote ±2 standard errors, displaying an approximate 95% confidence interval for our estimate.

**Table 1 materials-15-04729-t001:** Particle size distribution and apparent density of Ti6Al4V powder used in this study.

Description	Particle Size (µm)			Particle Size Distribution (µm)			Apparent Density (g/cm^3^)
	>53	≤53	<15	$D_{10}$	$D_{50}$	$D_{90}$	
Measured values (mass %)	1.3	96.7	2	21	37	51	2.38

**Table 2 materials-15-04729-t002:** Chemical composition of Ti6Al4V powder used in this study.

Element	C	O	N	H	Fe	Al	V	Ti
Standard values (mass %)	≤0.08	≤0.20	≤0.05	≤0.015	≤0.3	≤5.5–6.75	≤3.5–4.5	Balance
Measured values (mass %)	0.01	0.09	0.02	0.0022	0.22	6.44	4	Balance

**Table 3 materials-15-04729-t003:** Laser parameters used in manufacturing Ti6Al4V.

Parameter	Laser Power (W)	Layer Thickness (µm)	Scan Speed (mm/s)	Spot Size (µm)	Energy Density (J/mm^3^)	Hatch Distance (µm)
Value	190	30	1000	90	85	110

**Table 4 materials-15-04729-t004:** Standard deviations of individual measurements at different locations in the same slice (s1) and in different slices (s2), and means and standard deviations of the 20 slice means for each combination of structure type, variable and strut orientation. The standard deviations of the slice means were each based on 15 degrees of freedom, calculated using one-way ANOVA (slices within replicates) of the slice means. Additionally, the estimates are given with 95% confidence intervals of the excess (mean minus target) in microns and the excess as a percentage of the target value. All quantities are in microns, except for the last two columns, which are percentages.

			St.Dev.		Target Value	Slice Means		Excess (Mean–Target)		Percentage Excess	
			s1	s2		Mean	St.Dev	Estimate	(95% C.I)	Estimate	(95% C.I)
Diamond	ST	Vertical	18.5	21.0	300	307.5	10.52	7.5	(2.5, 12.5)	2.5	(0.8, 4.2)
		Horizontal	19.5	20.7	300	405.4	7.92	105.4	(101.6, 109.2)	35.1	(33.9, 36.4)
	PS	Vertical	18.9	21.9	1100	1066.9	11.51	−33.1	−(38.6, 27.6)	−3.0	−(3.5, 2.5)
		Horizontal	18.9	20.7	1100	1007.2	9.18	−92.8	−(97.2, 88.4)	−8.4	−(8.8, 8.0)
Gyroid	ST	Vertical	17.3	18.6	300	312.6	7.53	12.6	(9.0, 16.2)	4.2	(3.0, 5.4)
		Horizontal	37.1	37.3	300	498.6	7.79	198.6	(194.9, 202.3)	66.2	(65.0, 67.4)
	PS	Vertical	18.3	19.7	800	776.2	7.98	−23.8	−(27.6, 20.0)	−3.0	−(3.5, 2.5)
		Horizontal	33.8	35.3	800	631.3	11.83	−168.7	−(174.3, 163.1)	−21.1	−(21.8, 20.4)

## Data Availability

All data are contained within the article.

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
