# Peer review of "On the Morphological Deviation in Additive Manufacturing of Porous Ti6Al4V Scaffold: A Design Consideration"

_materials, 2022, doi:10.3390/ma15144729_

Round 1

Reviewer 1 Report

The manuscript entitled “materials-1786384” dealing with AM has been reviewed. The paper has been nicely written but needs significant improvement. Please follow my comments.

  1. Please add the powder information including packing density, distribution and size in a table.
  2. Add more quantitative results to the abstract.
  3. What is the novelty of this work? This should be highlighted in the abstract.
  4. How did you provide the process parameters in table 2? Based on manufacturer recommendation or your previous experience?
  5. Please provide a more thorough discussion for Figure 4.
  6. Additive manufacturing has many advantages over the conventional manufacturing method which can be highlighted in your paper. Please read the following article and add to the introduction to show the experimental application of additive manufacturing and the advantage of this process over conventional manufacturing like machining. “Additive manufacturing a powerful tool for the aerospace industry”
  7. Please update the introduction with the new publications in the field. Authors are encouraged to read and add the following new papers in the field.

·        High-cycle fatigue properties of curved-surface AlSi10Mg parts fabricated by powder bed fusion additive manufacturing

·        Proposal of design rules for improving the accuracy of selective laser melting (SLM) manufacturing using benchmarks parts

Author Response

Response to Reviewer 1:

  1. Please add the powder information including packing density, distribution and size in a table.

Response: Table.1 has been added to the updated manuscript in line 180 on page 4 which shows the particle size distribution and apparent density of Ti6Al4V powder used in this study.

  1. Add more quantitative results to the abstract.

Response: Yield strength of bulk sample, gyroid and diamond scaffolds plus the percentage excess of the strut thickness and pore size in vertical and horizontal struts were added to the abstract.

  1. What is the novelty of this work? This should be highlighted in the abstract.
    Response: Abstract has been updated to emphasis more on the novelty of this study, from line 19-24 on page 1.

  1. How did you provide the process parameters in table 2? Based on manufacturer recommendation or your previous experience?

Response: Laser parameters were provided by the manufacturer. They confirmed that the given parameters were optimised to gain the highest quality of print with least deviation between the as-designed and as-built scaffolds. Hence, the same suggested printing parameters were reported in this study.

  1. Please provide a more thorough discussion for Figure 4.
    Response: A paragraph was added from line 295-301 on page 8 to describe the surface indentation process and parameters including, maximum depth, contact depth, final depth and maximum load.

  1. Additive manufacturing has many advantages over the conventional manufacturing method which can be highlighted in your paper. Please read the following article and add to the introduction to show the experimental application of additive manufacturing and the advantage of this process over conventional manufacturing like machining. “Additive manufacturing a powerful tool for the aerospace industry”
    Response: Relevant content was extracted from the suggested paper and inserted in the introduction section from line 64-73 on page 2.

  1. Please update the introduction with the new publications in the field. Authors are encouraged to read and add the following new papers in the field.
  • High-cycle fatigue properties of curved-surface AlSi10Mg parts fabricated by powder bed fusion additive manufacturing

Response: Relevant content was extracted from the suggested paper and inserted in the introduction section from line 82-86 on page 2.

  • Proposal of design rules for improving the accuracy of selective laser melting (SLM) manufacturing using benchmarks parts.

Response: Relevant content was extracted from the suggested paper and inserted in the introduction section from line 142-148 on page 3.

Reviewer 2 Report

In this paper, the additive manufactured Ti6Al4V triply periodic minimal surface (TPMS) scaffolds with diamond and gyroid structures were characterized. The morphological and mechanical properties at macro and micro scales were examined.

Here are the suggestions before it could be accepted.

1. The literature should be updated, more literature should be in recent three years.

2. In the introduction, the disadvantages of the references should be summarized clearly to emphasize the importance of this work.

3. I do not know the meaning of Part 2.4. There is no description, no results. What is the meaing of “Fig. 6, 7, A.1, A.2, A.3, and A.4.”

4. There are too many formulas and symbols, so a nomenclature table is needed.

5. In fig.6, what is the meaning of the point with different color.

6. In part 3.1.2, where is “Figures A1 and A2”?

7. The conclusions should be refined, not list the value.

Author Response

The authors would like to thank the reviewers to provide feedback on this manuscript, many of which were positive and constructive. We have integrated these comments and suggestions into our revised manuscript. We summarized the feedback and listed our responses as enclosed for your reference.

I hope you will find satisfactory with the revised manuscript (we have highlighted the changes within the manuscript). Should you have further inquiry regarding our manuscript, please don’t hesitate to let us know.

Response to Reviewer 2:

  1. The literature should be updated, more literature should be in recent three years.
    Response: Introduction was updated with more literature in recent three years as highlighted. Lines 64-73 on page 2, lines 82-86 on page 2, lines 142-154 on pages 3-4.

  2. In the introduction, the disadvantages of the references should be summarized clearly to emphasize the importance of this work.
    Response: This has been added in the updated manuscript from line 158-160 on page 4.

  3. I do not know the meaning of Part 2.4. There is no description, no results. What is the meaning of “Fig. 6, 7, A.1, A.2, A.3, and A.4.”
    Response: This section is describing the software and method used to analyse the data. “Fig. 6, 7, A.1, A.2, A.3, and A.4.” was removed.

  4. There are too many formulas and symbols, so a nomenclature table is needed.
    Response: Nomenclature table has been added at the end of the manuscript. Line 781 on page 23.

  5. In fig.6, what is the meaning of the point with different color.
    Response: Each colour shows the replicate of the 5 replicates samples. Figure legend has been added to the figure and caption “Replicate 1, 2, 3, 4, 5 are shown by red, blue, purple, orange and turquoise color respectively” has been added to the figure caption.

  6. In part 3.1.2, where is “Figures A1 and A2”?
    Response: Both figures A1 and A2 (now changed to Figures S1 and S2) are shown in the Supplementary Material. I have now added this in lines between 375-376 to make it clear that these figures are in the Supplementary Material and not in the main text of this manuscript.

  7. The conclusions should be refined, not list the value.
    Response: Conclusion has been modified by removing some repeated results from results section. Point 5 was added to show how the data in this research paper can be useful for future work in this field. Lines 578-581 on page 19. 

Reviewer 3 Report

The manuscript is devoted to the study of the additive manufactured Ti6Al4V triple periodic minimum surface (TPMS) scaffolds with diamond and gyroid structures. In the manuscript, the authors studied in great detail the morphology, microstructure and mechanical properties of the obtained materials. Undoubtedly, the results of the work will be of interest to the scientific community and relevant for further research in the field of additive manufacturing.

I have a few small comments on the manuscript:

1.     The Materials and Methods part does not contain the dimensions of samples for determining the mechanical properties during tensile and compression tests.

2.     Figure 11. How did the authors differentiate between the regions of alpha and beta phases according to the depth of indentation? Figure 10 shows that all hardness marks fall on the alpha and beta phases. Thus, it is possible that the load-unload curves of the indentation correspond to areas with a greater or lesser volume fraction of the beta phase, due to which the indentation depth increases.

Author Response

The authors would like to thank the reviewers to provide feedback on this manuscript, many of which were positive and constructive. We have integrated these comments and suggestions into our revised manuscript. We summarized the feedback and listed our responses as enclosed for your reference.

I hope you will find satisfactory with the revised manuscript (we have highlighted the changes within the manuscript). Should you have further inquiry regarding our manuscript, please don’t hesitate to let us know.

Response to Reviewer 3:

  1. The Materials and Methods part does not contain the dimensions of samples for determining the mechanical properties during tensile and compression tests.
    Response: Length and width of all tested samples including the compression scaffold, compression bulk and tensile dogbone are shown in figures 2 and 3.

  2. Figure 11. How did the authors differentiate between the regions of alpha and beta phases according to the depth of indentation? Figure 10 shows that all hardness marks fall on the alpha and beta phases. Thus, it is possible that the load-unload curves of the indentation correspond to areas with a greater or lesser volume fraction of the beta phase, due to which the indentation depth increases.
    Response: The microstructure of α and β phases were not clearly visible under the digital microscope. Therefore, we used the results of load-displacement (P-h) graphs to understand where the indentations have been made, in an α or β phase. If we look at Figure 11, from left to right, the depth is increasing where smaller depths are corresponding to the indentations that are made into the α-phase, which are harder and higher depths are responsible for indentations that are made into the α+β phases or β-phases alone which are softer. This has also been reported in the literature as well. We have also compared hardness (Figure 12b) and Young’s modulus (Figure 12d) results of higher end α-phase and lower end β-phase with the literature and the results were similar; confirming our α-phase and β-phase.

Round 2

Reviewer 2 Report

It can be accepted